# Patient Activation, Depressive Symptoms, and Self-Rated Health: Care Management Intervention Effects among High-Need, Medically Complex Adults

**DOI:** 10.3390/ijerph18115690

**Published:** 2021-05-26

**Authors:** Cynthia F. Corbett, Kenn B. Daratha, Sterling McPherson, Crystal L. Smith, Michael S. Wiser, Brenda K. Vogrig, Sean M. Murphy, Roy Cantu, Dennis G. Dyck

**Affiliations:** 1College of Nursing, University of South Carolina, Columbia, SC 29208, USA; 2School of Nursing & Human Physiology, Gonzaga University and Providence Sacred Heart Medical Center, Spokane, WA 99258, USA; kenneth.daratha@providence.org; 3College of Medicine, Washington State University, Spokane, WA 992002, USA; sterling.mcpherson@wsu.edu (S.M.); crystal.lederhos@wsu.edu (C.L.S.); 4CHAS Health, Spokane, WA 99202, USA; mwiser@chas.org (M.S.W.); bvogrig@chas.org (B.K.V.); rcantu@chas.org (R.C.); 5Population Health Sciences, Weill Cornell Medicine, New York, NY 10065, USA; smm2010@med.cornell.edu; 6Department of Psychology, Washington State University, Spokane, WA 992002, USA; dyck@wsu.edu

**Keywords:** care management, high need, medically complex, chronic conditions, chronic illness, patient activation, depressive symptoms, depression

## Abstract

The purpose of this randomized controlled trial (*n* = 268) at a Federally Qualified Health Center was to evaluate the outcomes of a care management intervention versus an attention control telephone intervention on changes in patient activation, depressive symptoms and self-rated health among a population of high-need, medically complex adults. Both groups had similar, statistically significant improvements in patient activation and self-rated health. Both groups had significant reductions in depressive symptoms over time; however, the group who received the care management intervention had greater reductions in depressive symptoms. Participants in both study groups who had more depressive symptoms had lower activation at baseline and throughout the 12 month study. Findings suggest that patients in the high-need, medically complex population can realize improvements in patient activation, depressive symptoms, and health status perceptions even with a brief telephone intervention. The importance of treating depressive symptoms in patients with complex health conditions is highlighted.

## 1. Introduction

People who have complex physical health conditions that are further complicated by behavioral and/or social problems have been identified as a high-need, medically complex population [1,2]. These populations pose significant management challenges for healthcare systems. In the United States (U.S.), Federally Qualified Health Centers (FQHCs) have literally served the role of “safety net providers” for many patients in this population [3]. Following passage of the U.S. Patient Protection and Affordable Care Act in 2010, the number of people seeking care at FQHCs, including patients who were newly insured and part of the high-need, medically complex population, increased exponentially [4]. FQHCs and other health systems seek innovative methods of care, particularly for patients with highly complex needs.

The Chronic Care Model is widely employed as a framework for organizing and improving care and outcomes for people with chronic conditions [5]. It consists of six elements: (1) health systems and organization of healthcare; (2) decision support; (3) delivery system design; (4) clinical information systems; (5) self-management support; and (6) community resources (Improving chronic illness care, n.d.). The focus is on patient-centered care delivered by a proactive health team with activated patients and their families serving as members of the healthcare team. Medical homes are a strategy for providing patient-centered care to patients with complex care needs with the intent of increasing access and comprehensive and coordinated care to enhance productive patient and provider interactions and improve patient outcomes [6]. Many primary care practices, including FQHCs, have transitioned to a medical home model of care, often guided by the Chronic Care Model (Improving chronic illness care, n.d.). The Chronic Care Model provides a framework for delivering patient-centered care to people with complex care needs, while improving care processes, patient outcomes, and maintaining fiscal viability [5,7].

Clinics using the Chronic Care Model to guide practice improvement have had success in transforming their delivery system design, improving decision support, and implementing clinical information systems [8,9]. However, integrating patient self-management support such that patients can actively engage in care with their healthcare teams (i.e., be activated) has been more challenging [10,11,12]. Higher patient activation, defined as patients’ knowledge, skill, and confidence for managing their own health and healthcare, [13] has been associated with better self-management [14] and health, [15] and lower healthcare utilization [16,17]. Meticulous self-management and active engagement with healthcare providers is particularly important for patients who have multiple chronic conditions and complex care needs. Regrettably, most patients with multiple chronic conditions, particularly those who are socioeconomically disadvantaged, are poorly prepared to actively engage in self-management or have productive interactions with their healthcare teams [18,19,20]. Primary care providers and clinics have struggled to put processes in place to engage patients and enhance patients’ self-management skills due to resource limitations and barriers to transforming traditional care practices [14,17]. Patient activation is often sub-optimal and compromises the effectiveness of patient and provider interactions: a predicament that leads to sub-optimal health outcomes [14,15,16,17]. Hence, providing self-management support to improve patient activation may be particularly critical for patients with multiple chronic conditions.

Care management, a self-management support strategy to help patients navigate the health system and engage in their own care (i.e., improve patient activation), is particularly appropriate for patients with multiple chronic conditions because it shifts attention from disease management that is focused on meeting standardized clinical targets, to a tailored approach that is based on patients’ needs and preferences [21]. The primary purpose of this clinical trial was to compare the outcomes of a chronic care management intervention (CCMI) versus a brief attention control telephone intervention (ACTI) on changes in patient activation among adults in the high-need, medically complex population. The CCMI was selected because it is an evidence-based approach shown to be effective in a prior demonstration project in Washington state [21]. The ACTI was chosen to provide a brief intervention that may offer some benefit to participants who were randomized to the control intervention. The research team believed that offering a brief intervention was more ethical among a population of high-need, medically complex patients as compared to offering only standard care, and that it would also be useful for engaging and retaining patients who were randomized to the control arm of the study. In addition to evaluating intervention effects on patient activation, intervention effects were evaluated for two other self-reported outcomes, depressive symptoms and self-rated health, both known to co-vary with patient activation [22]. Participants had multiple chronic conditions and received primary care at an FQHC with a medical home delivery model.

## 2. Methods

### 2.1. Design

A prospective, single-blind clinical trial with participants recruited from an FQHC, randomized to a CCMI or an ACTI, was completed. Investigators and the study coordinator who assessed outcome measures were blinded to group assignment. Due to the behavioral nature of the study interventions, participants were not blinded to assignment. The study procedures were approved for human subject participation by the university’s institutional review board, and the study was registered on clinicaltrials.gov (NCT02136732).

### 2.2. Recruitment and Randomization

Patients who were 45 years of age or older, received primary care at the participating FQHC, had two or more chronic conditions (physical, psychological, and/or behavioral), and two or more acute care encounters, at least one of which was a hospitalization, in the previous 12 months were identified through a report generated from the FQHC’s electronic health record. Eligibility verification was first completed by health information system personnel and then confirmed by patients’ primary care providers. The study coordinator telephoned fully screened patients, and for those interested, scheduled an in-person meeting to further explain the study and to obtain written informed consent and baseline measures.

A total of 5168 patients were screened, 1108 met the inclusion criteria, and 290 (26% of those meeting inclusion criteria) were enrolled in the trial. The sample size was based on statistical power calculations for enhancing patient activation for self-management (as measured by the Patient Activation Measure). Previous research demonstrated small effect sizes (0.20 standard deviation units) for chronic condition self-management support and coaching being associated with enhanced activation as compared to standard care [23,24,25]. We conservatively assumed a 25% attrition rate based on our previous experience conducting longitudinal research with similar populations, which resulted in an a priori power analysis sample size of 224 (*n* = 112 per group). Thus, our study was well powered for detecting changes in patient activation. Consenting participants were randomly assigned to CCMI or ACTI groups using the URN randomization procedure [26]. Randomization was stratified by age and gender. Both study groups continued to receive usual care from the FQHC medical home as well as other health and community services. A total of 22 participants were disenrolled from the intervention due to ineligibility (e.g., moved out of area, elected to participate in an insurance-sponsored case management program) allowing 268 eligible for analysis (Figure 1).

### 2.3. Data Collection

The primary study outcome was patient activation, defined as people’s knowledge, confidence, and skill to participate in self-management and engage with health professionals as an active member of their own health team [26]. Patient activation was measured by the Patient Activation Measure (PAM-13), which has 13 Likert response questions that range from strongly disagree to strongly agree. The PAM-13 has good reliability [13] and demonstrated validity as higher (better) PAM-13 scores have been associated with improved outcomes for adults with chronic conditions [17,27] and lower use of health services [15,28,29]. The PAM-13 was administered to both study groups at baseline, 3, 6, 9, and 12 months. PAM-13 item ratings were recorded on an electronic worksheet provided by Insignia Health^®^, which calculates PAM-13 scores for each participant. Higher scores correspond to higher activation. People who have the lowest scores (<47) are characterized as being passive, lacking confidence in their ability to manage their health, and having low adherence to healthcare professionals’ recommendations [30]. Those who have the highest PAM-13 scores are people who can maintain self-management behaviors and interact as a partner with their healthcare team [30].

Self-rated health was measured using a 1-item measure whereby participants rated their current health as one of five categories ranging from very poor to very good at baseline and every subsequent three months [31]. Self-rated health has been shown to influence healthcare utilization [32] and mortality [33]. The 9-item Patient Health Questionnaire (PHQ-9) is a reliable and valid depression screening instrument that is widely used in clinical care, including at the FQHC where patients in this study received care [34,35]. The PHQ-9 was administered to assess depressive symptoms at baseline and 12 months. Higher scores are associated with more depressive symptoms (5–9 mild, 10–14 moderate, and 15+ severe). Sociodemographic variables collected at baseline included sex, race, marital status, type(s) of health insurance, educational attainment, and health literacy. Health literacy was measured using the Rapid Assessment of Adult Literacy in Medicine—Revised (REALM-R) [36].

### 2.4. Intervention and Protocols

#### 2.4.1. Treatment Group Procedures

The CCMI was based on a successful demonstration project in Washington state [21] and delivered by an intervention-trained team of a social worker (SW) and a registered nurse (RN). The SW completed a comprehensive bio-psychosocial assessment on each participant, including screening for depressive symptoms. The RN completed a health history, nursing assessment, and documented the medicines participants reported taking. Based on the RN and SW assessments, including participants’ baseline patient activation scores, a Health Action Plan (HAP) was developed collaboratively between each participant and the RN/SW team. The goal-directed individualized plan focused on (1) self-management, (2) appropriate use of medical, social services, and other community resources, (3) healthcare system navigation, and (4) collaboration with the healthcare team to achieve self-identified health goals (see Appendix A for exemplar case).

As the CCMI continued, ongoing visits and telephone calls supported participants as they developed skills to achieve their HAP goals and skills to engage as partners with their healthcare team. The RN and SW used motivational interviewing techniques to help participants to initiate HAP goals, to facilitate participants’ abilities to reach their HAP goals, and to increase participants’ abilities to have productive interactions with their healthcare team. For the latter, participants were encouraged and taught how to plan relevant questions for their healthcare team as well as to implement techniques to advocate for themselves. Throughout the 12 month study the RN and SW used teach-back methods and role playing to promote new skill acquisition among participants. The RN and SW evaluated participants’ responses to the HAP, modifying it in collaboration with participants as goals were met. They also monitored social services and facilitated connection to other resources as needed and agreed upon by the participant.

Participants randomized to the CCMI received a minimum of one visit and one telephone call per month for the 12 month intervention. Most CCMI visits were conducted in participants’ homes, which has been shown to be an effective care management delivery model for people with multiple chronic conditions [37]. A few participants preferred visits be conducted at a community location such as a coffee shop or the FQHC clinic. When indicated by the HAP and invited by the participant, the RN and SW also accompanied participants to primary or specialty care outpatient clinic visits. Further, on rare occasions, participants who were hospitalized were visited by the RN or SW while they were in the hospital.

#### 2.4.2. Attention Control Group Procedures

The ACTI was delivered by a protocol-trained Social Services Assistant (SSA). The SSA scheduled an initial home visit with participants to individually orientate each participant to the ACTI procedures and complete baseline assessments. To deliver the ACTI at the initial visit and during subsequent monthly telephone calls, which generally lasted from one to two minutes, the SSA asked participants if they had needs that could be met by available community resources. Participants could self-identify needs and ask to address any desired need. If the participant indicated one or more needs, the SSA sent information to participants through the United States Postal Service to enable participants to initiate contact with the community resource.

#### 2.4.3. Intervention Fidelity

Fidelity to the interventions and to outcome assessments was monitored qualitatively and quantitatively. At the outset of the study, we were concerned that it may be difficult to engage with participants at the desired frequency to effectively deliver the CCMI and ACTI. Thus, a portion of the fidelity protocol for the 12 month study was an established frequency for participant contacts each month. We tracked participant contacts, and overall adherence to the protocol for participants contact was 91%. The 134 participants in the CCMI group had 8242 contact attempts and 5167 successful contacts (63%) with the RN and SW for an average of 62 contacts per participant during the 12 month study. In the ACTI group (*n* = 134 participants), 4445 total contacts were attempted, and 2004 contacts were completed (45%) for an average of 15 contacts per participant during the 12 month study.

We also had a desire to consistently implement study interventions with the participants. To attain that goal, we qualitatively monitored intervention fidelity every 3 months. The primary investigator or a co-investigator observed the RN, SW, and SSA during select participant visits or listened to select telephone call interactions with participants. Fidelity was rated from 1 (low) to 5 (high) using a process evaluation checklist that was based on the study protocols. Specifically, visits and the intervention delivery style of the RN, SW or SSA were evaluated on a scale of 1–5 for collaboration and for supporting the participants’ autonomy. The low value (1) on the collaboration scale corresponded to: *Clinician actively assumes the expert role during most of the interaction with the participant. Collaboration is absent.* The high value (5) on the collaboration scale was: *Clinician actively fosters and encourages power sharing during the interaction so that the participants’ ideas substantially influence the activities and outcomes of the interaction.* For the autonomy/support scale, the low value (1) corresponded to an interaction in which the *Clinician actively detracts from the participant’s choice or sense of control* whereas the high value (5) depicted an interaction whereby the *Clinician adds significantly to the participant’s expression of autonomy to markedly expand the participant’s feeling and experience of choice and control.* Activities commonly performed during home visits or telephone calls were customized for the RN, SW, and SSA. Of note is that the protocol called for the SSA to only provide information about community services by sending information through the U.S. Postal Service. To try to clearly differentiate the CCMI from the ACTI, ACTI protocols directed the SSA not to provide teaching, self-management support, or verbal information about community resources to participants. Fidelity monitoring of visits and phone calls, completed by the primary investigator or one of the co-investigators who had this function as part of her research role, were rated for person-centered intervention delivery, and written comments were provided to justify the ratings. Results were shared with the clinicians following the visit/telephone call to improve and standardize interventions. Using this process, fidelity to the respective protocols averaged over 4.5 on the 5.0 scale for all personnel.

### 2.5. Statistical Analysis

Differences between treatment groups, and possible interaction effects between group and pre-specified covariates were evaluated using inferential statistical analyses. Basic descriptive analyses evaluated the means and standard deviations, and the counts and frequencies for each of the study variables. Inferential analyses to evaluate the effect of CCMI and ACTI assignment on the PAM-13 scores were conducted with random-effects regression. Missing data were handled in a manner consistent with expert recommendations wherein maximum likelihood was utilized, assuming data were “missing at random” [38,39]. The multi-level modeling framework included main effects for treatment group, time, and the treatment group by time interaction. Baseline depressive symptoms were included as a covariate because the CCMI group had higher depressive symptomatology despite randomization, revealed during the descriptive phase of analyses. Sub-group analyses based on age were conducted as planned a priori. In addition, exploratory analysis of health status as a covariate was conducted because it has been previously shown to be associated with the PAM-13 [40]. All analyses were conducted using Stata 14.0 (StataCorp LLC, College Station, TX, USA).

## 3. Results

Participants (*n* = 268) were 64% female, mean age of 55 years (SD = 7 years), and 85% were Caucasian, which reflected the community’s racial composition. Medicaid was the primary type of insurance (73%), and most participants had at least a high school education (88%). Types of chronic health conditions were similar between groups, with hypertension, hyperlipidemia, diabetes, depression, and substance-use disorders being the most common diagnoses. Demographic characteristics and baseline measures of patient activation, self-rated health, and depressive symptoms of participants in the CCMI and ACTI groups are reported in Table 1.

Groups did not differ at baseline for any of the above variables (*p* < 0.05, two-tailed) using independent samples t-tests (continuous variables) and X2 tests of independence (categorical variables), with the exception of PHQ-9. There was a significant difference in baseline PHQ-9 scores between the PCPI and ACI groups t (237) = −2.94, *p* < 0.05, such that the PCPI group had significantly higher scores, indicating higher levels of symptoms of depression.

Significant differences in PAM-13 scores were observed over time in both the CCMI group and ACTI group, (β = 1.04; 95% CI = 0.62–1.45, *p* < 0.01), but there was no difference between the two groups (β = 0.64; 95% CI = −0.19–1.47, *p >* 0.05), and there was no statistically significant interaction between treatment group and time. Self-rated health significantly increased over time for both treatment groups (β = 0.06; 95% CI = 0.03–0.09, *p* < 0.01). No group differences were observed in self-rated health (*p* > 0.05). Depressive symptoms declined in both groups over time (β = −2.09; 95% CI = −2.87–−1.31, *p* < 0.01), with greater improvement observed in the CCMI group (β = −1.74; 95% CI = −3.28–−0.19, *p* < 0.05).

Older participants in the CCMI group had significantly greater improvements in PAM-13 scores as compared with older participants in the ACTI group (i.e., years of age; β = 0.47; 95% CI = 0.11–0.82, *p* < 0.05). Similarly, older participants who received the CCMI had better self-rated health over time than older participants in the ACTI group (β = 0.01; 95% CI = 0.00–0.03, *p* < 0.01). While depressive symptoms improved in the CCMI group as compared to the ACTI group, the differences were more pronounced and statistically significant among older participants (β = −0.14; 95% CI = −0.26–−0.02, *p* < 0.05). In addition, higher baseline depressive symptoms were associated with lower PAM-13 scores throughout the trial in both ACTI and CCMI group participants (β = 0.84; 95% CI = 0.83–0.92, *p* < 0.01). Table 2 reports mean changes over time and between groups for measures of PAM-13, health status rating, and PHQ-9.

## 4. Discussion

We evaluated the effects of a chronic care management intervention compared to a brief telephone intervention on patient activation, within a high-need, medically complex patient population who had multiple chronic conditions. Both groups had baseline patient activation scores that placed them in the middle range of activation, indicating they were beginning to build self-management skills and were striving to become goal-oriented and meet best practices [30]. Both groups achieved modest, yet statistically significant, increases in patient activation. Those who received minimal telephone contact increased their activation similarly to those who received the more intensive in-person intervention. However, older participants who received the CCMI had higher activation scores over time compared to those in the ACTI group. These finding may suggest that within-patient activation can be improved with minimal extra contact with health system professionals over time. Further, there was an age-based intervention response for patient activation, with older participants showing greater improvements in patient activation than younger participants. 

Several investigators have reported no treatment group differences when implementing interventions to improve patient activation (measured by PAM-13) among patients with complex medical and psychosocial care needs [41,42,43,44,45,46,47,48,49]. Kangovi and colleagues conducted a similar randomized clinical trial that evaluated the effects of goal-setting, or goal-setting combined with 6 months of community health worker support [46]. Participants had multiple chronic conditions, lived in high poverty neighborhoods, and were uninsured or publicly insured. After the trial, the two groups did not differ in patient activation; however, the group that received community health worker support had improvements in mental health and disease self-management, as compared to those who had only the goal-setting intervention. In another 6 month, cluster-randomized clinic-based intervention that provided tailored self-management support for patients with chronic conditions, participants receiving the active intervention had improvements in self-monitoring but no improvements in patient activation [42]. Similarly, Hudon and colleagues implemented a 6 month pragmatic study that implemented nurse case management and a self-management intervention for patients with multiple chronic conditions and social vulnerability [45]. There were no significant improvements in patient activation from pre- to post-intervention [45]. Improving activation in patients with complex physical, behavioral, and social needs seems to be particularly challenging [41,42,43,44,45,46,47,48,49,50]. However, in our study, the CCMI and the ACI groups realized higher patient activation over time, but the differences between groups were only significant for older participants. By contrast, a single group, 12 month, primary-care-based intervention study in Australia found that older age was associated with lower PAM-13 scores over time [51]. In John et al.’s study, approximately half of the participants were over age 70, whereas in our study, very few participants were over age 70 [51]. Age and patient activation may have a curvilinear relationship such that after a certain age, activation levels decline.

Our findings among high-need, medically complex patients align with the recommendations of others that integrated care should address not only patients’ physical and behavioral needs but also basic needs, such as housing and food security. Patients social care needs must be met to also achieve substantive improvements in patient activation and other health-related outcomes [52,53,54,55,56,57,58,59]. Results from our study suggest a need to evaluate whether self-management support that extends beyond one year can positively affect mental, physical, and social health and sustain increases in patient activation for high-need, medically complex patient populations across the age spectrum.

Higher self-rated health has been shown to be associated with better patient activation [22]. Participants in both the CCMI and ACTI groups had improvements in health status rating, with no significant difference between groups, except patients in the CCMI group who were older in age had higher self-rated health over time than older patients in the ACTI group. Kangovi and colleagues reported similar increases in self-rated health in their control and intervention groups in a population with complex chronic conditions who were recruited from primary care clinics [46].

A high level of depressive symptoms was present among the participants in the current study, with the CCMI group having higher depressive symptoms than the ACTI participants at baseline. Consequently, the depressive symptom score was used as a covariate in our regression models. However, we also found that higher depressive symptoms were associated with lower patient activation throughout the 12 month study. An inverse relationship between depressive symptoms and patient activation has been a finding in multiple other studies [15,17,22,44,46,53,60,61]. Sacks and colleagues reported that among patients with moderate to severe depression, higher depression severity, lower treatment response rates, and lower remission rates after one year were associated with lower levels of patient activation [61]. In a randomized controlled trial that tested a depression self-care intervention, the 6 month intervention did not improve patient activation [62]. In our study, whereby the RN, SW, and SWA who delivered the interventions had knowledge of participants’ baseline levels of depressive symptoms, both groups had significant reductions in depressive symptoms and significant increases in patient activation over time. The reduction in depressive symptoms was greater in the CCMI group over time as compared with the ACTI group, even when controlling for baseline levels of depressive symptoms. Thus, we may cautiously conclude that the CCMI intervention had a positive impact on depressive symptoms, particularly for those who were older in age. Based on literature showing an inverse relationship between patient activation and depressive symptoms, reducing depressive symptoms may be necessary to improve patient activation—a recommendation made 13 years ago with seemingly little uptake [23].

### 4.1. Limitations and Strengths

Some limitations should be considered when interpreting the results of this study. The trial was conducted with patients from multiple clinics within a single FQHC in the northwestern United States; therefore, findings may not be generalizable to other areas of the country, particularly those with greater racial and ethnic diversity. People who chose to participate in the study may have had different characteristics as compared to those who declined participation. Participants in the CCMI group had higher levels of depressive symptoms, which may have influenced the effectiveness of the intensive intervention. However, CCMI participants also had greater improvements in depressive symptoms compared to ACTI participants, suggesting the intervention may have positively influenced mental health. The finding that patients in both groups had improvements in patient activation, depressive symptoms, and self-rated health may reflect a regression to the mean, a validity threat observed in similar studies that recruited patients with high acute care utilization [63]. However, several characteristics of our study reduced the chances of regression to the mean. Participants were randomly assigned to treatment groups, and the study coordinator who assessed the outcomes was blinded to assignment [64]. Further, our inclusion criteria for high utilization was only two or more acute care encounters in the past 12 months, and the timing of participants’ most recent acute care encounter prior to study entry varied. Additionally, some participants had acute care encounters during the study; thus changes in measures over time cannot be attributable to participants being more acutely ill at the time of the baseline measures versus at the subsequent measurement times [63], which would be the most logical reason to observe regression to the mean in this study. The inclusion criteria of only two or more acute care encounters in 12 months, one of which was a hospitalization, may have also been a limitation of the study in that it is possible that some participants who met the criteria did not have complex, high-need chronic conditions. For instance, they could have had low need chronic conditions (e.g., osteoarthritis and chronic migraine headaches) and visited the emergency department once for headache management and had elective hip surgery. However, primary care providers had to “approve” their patients’ suitability for the study prior to recruitment; therefore, while that scenario is possible, we believe it is unlikely.

Strengths of the study included that it was more pragmatic than many clinical trials because eligibility criteria were broad (e.g., no limitations were placed on diagnoses, participant age range was wide (45 years and older), the intervention was tailored to participants’ individual health goals and needs, intervention fidelity was routinely evaluated, and the interventions were delivered across multiple care settings for 12 months [65]. Thus, our study addressed many of the recommendations for improving care to the high-need, medically complex population: a rigorous, person-centered tailored intervention and frequent engagement between participants and care providers across care settings, [54,63,66,67]. Baseline depressive symptoms, patient activation scores, and self-rated health were known to the nurse and social worker who implemented the CCMI, which helped them individualize and adjust interventions over time (see Appendix A for exemplar of CCMI process). Consequently, the study’s context more closely mirrored actual clinical situations faced by FQHC clinician teams who care for patients with multiple chronic conditions.

### 4.2. Implications

More research as to the role of patient activation, depressive symptoms, self-management, and health outcomes in populations with complex chronic conditions is warranted. Our study findings support the need for integrative care that addresses patients’ behavioral, social, and physical health. Patients who fall into the high-need, medically complex group with multiple chronic conditions may require long-term support from health professionals and/or paraprofessionals to attain and sustain improvements. Research implications include the need to test the effects of high-intensity and low-intensity longitudinal interventions, possibly using a tailored adaptive intervention based on risk and age, for patients who are members of high-need, medically complex populations. Some patients in this population may require access to community resources to stabilize social contexts such as housing and food security. In addition, care management support that extends two or more years may be required to enable some members of the high-need, medically complex population to attain skills to be activated and engaged members of their own healthcare team [63,66].

## 5. Conclusions

In summary, participants in both groups had improvements in patient activation, self-rated health, and depressive symptoms over time. This finding is encouraging because it may suggest that patients with complex chronic health problems respond favorably to interventions, including a brief, low-cost telephone intervention, across several self-reported domains. However, participants in our study who were older and received the more intensive, in-person intervention significantly improved levels of each outcome as compared to participants who received the brief telephone intervention. Participants of all ages who received the intensive in-person intervention delivered by a nurse and social worker who had knowledge of their baseline depressive symptoms scores had greater improvements in depressive symptoms than those who received the brief telephone intervention. Patient activation was consistently lower over time among participants in both groups when baseline depressive symptoms were higher. Our findings support the work of others in noting the criticality of managing depressive symptoms to improve patient activation and overall health outcomes in patients with complex chronic conditions.

## Figures and Tables

**Figure 1 ijerph-18-05690-f001:**
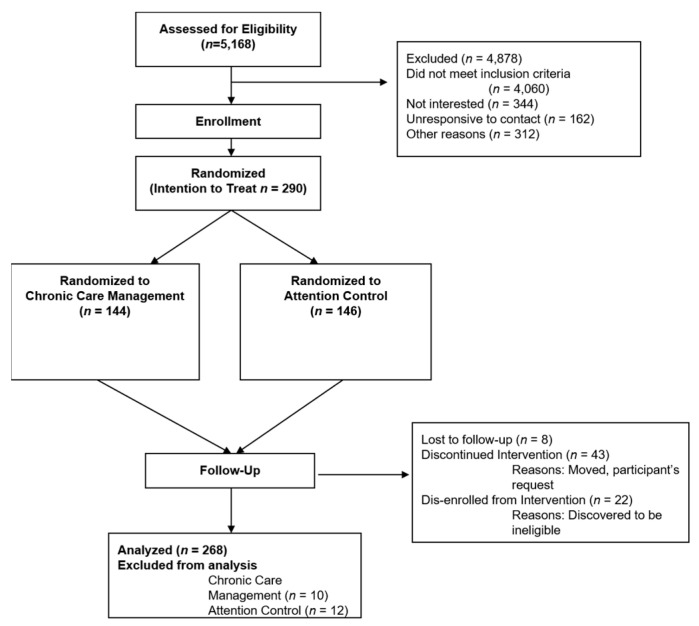
CONSORT diagram of Chronic Care Management randomized controlled trial.

**Table 1 ijerph-18-05690-t001:** Participants’ baseline characteristics.

		Chronic Care Management Intervention (CCMI)	Attention Control Intervention (ACI)
Baseline characteristics		(*n* = 144)	(*n* = 146)
Age, M (SD)		55.35 (6.94)	55.24 (7.21)
Sex			
	Male (%)	53 (36.8%)	52 (35.6%)
	Female (%)	91 (63.2%)	94 (64.4%)
Not Latino		142 (98.6%)	143 (97.9%)
Race			
	Caucasian (%)	124 (86.1%)	122 (83.6%)
	Black (%)	7 (4.9%)	10 (6.8%)
	American Indian (%)		3 (2.1%)
	Asian (%)	1 (.7%)	
	Other (%)	12 (8.3%)	11 (7.5%)
Marital Status			
	Single	23 (17.3%)	27 (20.5%)
	Committed/Married	46 (34.6%)	39 (29.5%)
	Divorced/Separated	55 (41.4%)	53 (40.2%)
	Single after death of spouse	9 (6.8%)	13 (9.8%)
Insurance			
	Medicare	32 (22.5%)	25 (17.4%)
	Medicaid	93 (65.5%)	84 (58.3%)
	Dual Medicare/Medicaid	10 (7.0%)	21 (14.6%)
	Other public	1 (0.7%)	1 (0.7%)
	Commercial/HMO	5 (3.5%)	8 (5.6%)
	Self-pay	1 (0.7%)	5 (3.5%)
Education			
	Some high school	15 (11.5%)	16 (12.2%)
	High school	34 (26.2%)	44 (33.6%)
	Some college/technical	55 (42.3%)	43 (32.8%)
	2-year college graduate	13 (10.0%)	16 (12.2%)
	4-year college graduate	10 (7.7%)	10 (7.6%)
	Some post bac or greater	3 (2.3%)	2 (1.6%)
Health literacy M (SD)		7.0 (2.1)	7.1 (1.9)
Patient Activation Measure Score M (SD)		55.9 (13.5)	56.6 (11.6)
Health Status Rating: *n* (%) at risk		138 (95.8%)	139 (95.2%)
Patient Health Questionnaire (PHQ-9)		11.5 (6.5)	9.31 (5.1)

**Table 2 ijerph-18-05690-t002:** Means and standard deviations (in parentheses) of key, clinical trial measures across time and treatment group.

Measure	Group	Baseline	3 Months	6 Months	9 Months	12 Months
**PAM-13**	CCMI	55.9 (13.5)	58.2 (14.7)	60.6 (14.5)	58.4 (13.0)	61.7 (14.6)
	ACTI	56.6 (11.6)	59.0 (12.6)	58.4 (11.7)	59.2 (12.3)	60.5 (12.7)
**Self-Rated Health**	CCMI	1.9 (0.8)	2.3 (0.9)	2.2 (0.9)	2.2 (0.8)	2.4 (0.9)
	ACTI	2.0 (0.9)	2.2 (0.8)	2.2 (0.9)	2.3 (0.8)	2.2 (0.9)
**Depressive Symptoms**	CCMI	11.5 (6.5)				8.4 (5.5)
	ACTI	9.3 (5.0)				8.0 (5.5)

## Data Availability

Contact the primary investigator at corbett@sc.edu to request de-identified data supporting the reported results.

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
