# Peer review of "Patient Activation, Depressive Symptoms, and Self-Rated Health: Care Management Intervention Effects among High-Need, Medically Complex Adults"

_ijerph, 2021, doi:10.3390/ijerph18115690_

Round 1

Reviewer 1 Report

Some background information on how the Chronic Care Model may influence the different health outcomes is missing from the introduction. Further I believe some constructs should be briefly outlined and explained. For example how did the authors define “patient activation” or what conceptualization for this construct was used?

Methods: Sample size calculation seems not to have been done? If so, based on what outcome? Intervention description could benefit from more details so readers have a better understanding of it.

Results: regression results could be presented in table form.

Discussion: high-needs and hi-cost patient population seems out of the blue. There is no evidence to support that your population was such. Do you have any data on health care utilization or costs of care in the patient group? Which chronic disease was most prevalent?

Reviewer 2 Report

Thank you for the opportunity to review this paper. The study is a randomized controlled trial that compared the outcomes (i.e., patient activation, depressive symptoms, and self-rated health) of high-need, medically complex adults in two intervention groups (i.e., care management intervention and attention control telephone intervention) over a 12-month period. Results showed that both groups demonstrated significant improvements in patient activation and reductions in depressive symptoms; however, participants that received the care management intervention reported greater reductions in depressive symptoms. I believe this study provides some helpful information that adds to the literature on intervention for the management of depression among adult patients with complex health needs. However, I do have the following feedback/questions for the authors:

It is not clear why these two particular interventions (i.e., CCMI and ACTI) have been selected for comparison. The introduction should be expanded to provide a rationale for this.

Patients were recruited if they were 45 years or older, received primary care at the FQHC, had 2 or more chronic conditions, and 2 or more acute care encounters in the last 12 months. There are many chronic conditions that relatively minor or fail to significantly impact an individual’s functioning and quality of life. Acute care encounters may be non-emergency, result from relatively minor issues, or do not appear to be related to any established health condition (e.g., a man has arthritis and ends up in urgent care for twisting an ankle while jogging). Did your inclusion criteria account for these potential issues? If not, how might this limit the interpretation of your findings?

Frequency of participant contacts was used as a measure of implementation fidelity, but this seems rather insufficient. What is the justification for qualifying frequency of contacts as an appropriate measure of fidelity?

Fidelity does not only apply to participants. What steps, if any, were taken to ensure that the intervention was delivered with fidelity? If none, then some explanation of this as a limitation seems warranted.

Evaluation of fidelity included a brief mention of qualitative analyses, but there is no description of what this involved and the results of these analyses.

For the attention control group, what types of community resources were offered to patients? And were resources tied to what was available in the community in which the patient resides? If so, how might that affect the types of resources provided and resulting contact rates?

Round 2

Reviewer 1 Report

I thank the authors for the changes done, I have no further comments.

Reviewer 2 Report

The authors have made sufficient edits to the content of the manuscript and the major concerns I reported in my review of the initial submission have been adequately addressed. I am delighted to see that the revised version has expanded on several sections of the paper that provide further clarifying details about the study's methods and limitations.